# Remediation of Heavy Metals in Polluted Water by Immobilized Algae: Current Applications and Future Perspectives

Zhonghao Chen [1], Ahmed I. Osman [2,*], David W. Rooney [2], Wen-Da Oh [3] and Pow-Seng Yap [1,*]

[1]  Department of Civil Engineering, Xi'an Jiaotong-Liverpool University, Suzhou 215123, China; zhonghao.chen21@student.xjtlu.edu.cn
[2]  School of Chemistry and Chemical Engineering, Queen's University Belfast, Belfast BT9 5AG, Northern Ireland, UK; d.rooney@qub.ac.uk
[3]  School of Chemical Sciences, Universiti Sains Malaysia, Gelugor 11800, Penang, Malaysia; ohwenda@usm.my
*  Correspondence: aosmanahmed01@qub.ac.uk (A.I.O.); powseng.yap@xjtlu.edu.cn (P.-S.Y.)

**Abstract:** The progression of urban industrialization releases large quantities of heavy metals into water, resulting in the severe heavy metal contamination of the aquatic environment. Traditional methods for removing heavy metals from wastewater generally have varying removal efficiencies, whereas algae adsorption technology is a cost-effective and sustainable bioremediation technique. A green technology that immobilizes algae through a carrier to improve biosorbent's stability and adsorption performance is immobilization technology. The purpose of this review is to study the optimization strategy of the immobilization of algae for the bioremediation of heavy metals and to comprehensively analyze immobilized algae technology in terms of sustainability. The analysis of the mechanism of heavy metal removal by immobilized algae and the parameters affecting the efficiency of the biosorbent, as well as the approach based on life cycle assessment and economic analysis, allowed the identification of the optimization of the adsorption performance of immobilized algae. This provides a theoretical basis for the practical application of algal bioremediation.

**Keywords:** water remediation; immobilized algae; biosorption; heavy metals; life cycle assessment

## 1. Introduction

Urbanization, industrialization, and intensive human activities have contributed to increasing occurrences of water pollution. Due to the rapid expansion of industry and human activity, heavy metals, which are naturally non-biodegradable and non-thermally degradable, have been released into the environment. Residual heavy metal ions from residential and commercial effluent have contaminated rivers, lakes, and oceans. High concentrations of cadmium and copper were detected in well water near the Copper Mountain mine in China's Hubei province [1]. Heavy metal stress endangers crops by reducing plant growth and yield parameters [2]. Heavy metals reduce the biodegradability of organic pollutants, which prolongs their environmental persistence and exacerbates the effects of other hazardous wastes [3]. Due to the non-essential nature of these metal ions, they are highly mobile in aquatic systems and are regarded as extremely hazardous even in trace amounts [4]. Humans may be exposed to heavy metal ions via biosorption and accumulation in the aquatic food chain, resulting in severe health problems, organ tissue damage, and cell deterioration due to excessive ingestion [5].

Chemical precipitation [6], adsorption [7–10], electrodialysis [11], ion exchange [12], membrane filtration [13], coagulation [14]/flocculation [15], and electrochemical precipitation [16] are some of the currently available methods used to treat heavy metals in polluted water. Each of these methods has its own advantages and limitations. In view of advocating environmental sustainability, it is of great interest to develop innovative, eco-friendly and sustainable methods which can ensure the complete removal of the heavy metals [17,18].

An environmentally friendly technology is bioremediation, which uses the metabolic potential of microorganisms to remove heavy metals through a series of physicochemical interactions which occur between the functional groups of microorganisms and the heavy metals [19]. Biosorption is among the most desirable methods for removing radioactive ions and heavy metals from wastewater. In addition to being cost-effective, biosorption offers the possibility of recycling waste materials [20]. Algae are natural biomass and they have varying degrees of affinities for heavy metals [21]. Algal biosorption makes heavy metal ion removal from wastewater more cost-effective, efficient, and environmentally safe [22]. Due to their capacity for nutrient removal, adsorption, and regenerative and sustainable nature, algae are utilized to treat a variety of industrial, agricultural, and mining wastewaters, as opposed to conventional treatment systems [23]. Algal growth in aquatic systems is an eco-friendly, greener, and sustainable bioremediation technique, because of its photosynthetic capacity to absorb $CO_2$ and its adaptability to grow in different types of wastewaters [24].

Due to the ability of algal cells to attach to specific surfaces, biomembranes are produced naturally, thereby reducing the overall cost of separating algal biomass from the rest of the medium [25]. Using larger immobilized microalgae beads or carriers can make the typical harvesting and dewatering process more straightforward and energy efficient. Because microbeads may prevent the release of immobilized microorganisms into wastewater, immobilized algal cultures may also mitigate the disruption of the native ecosystem caused by the introduction of foreign microorganisms [26].

Currently, heavy metal removal technologies for wastewater treatment have yet to fully transition to algae remediation technologies. Free algae are applied to actual complex wastewater environments where there is cell leakage and cannot be highly selected for the low concentrations of heavy metals. Furthermore, immobilized algae technology is not widely applied in the actual production industry, and the feasibility of immobilized algae to remove heavy metals from wastewater with high efficiency needs further verification. The complex environment with multiple influencing parameters is an important aspect of the immobilized algae application. Moreover, to date, it appears that there are no well-documented studies on life cycle assessment or the economic analysis of immobilized algae for wastewater treatment, and thus these studies merit attention and investigation. The review aims to analyze the feasibility and sustainability of immobilized technology algae in removing heavy metals from the aqueous environment and to make recommendations for optimizing the performance of immobilized algae. In this review article, the mechanisms of heavy metals removal by immobilized algae were analyzed. In addition, the properties of various immobilization strategies including adsorption, encapsulation, entrapment, and self-immobilization, were summarized. Moreover, the effects of parameters of immobilized algae for the removal of heavy metals were analyzed. Lastly, the sustainability of immobilized algae based on life cycle assessment and economic analysis was reviewed.

## 2. Immobilized Algae Bioremediation Technology

### 2.1. Methodology

In order to accurately review the current mechanisms of heavy metal removal by immobilized algae and the approaches of algae immobilization, in this review, we searched for studies in the Web of Science (WoS) and Scopus databases and further identified keywords concerning topics that fit this study: algae, biosorption, bioaccumulation, biotransformation, adsorption, encapsulation, trapping, and self-immobilization. The time range was adjusted to 2016–2023 based on the filtering results, thus identifying the most recent research papers on the bioremediation of heavy metals by immobilized algae. With the information obtained from the data of the papers, the mechanism of heavy metal biosorption by algae and the method of immobilized algae were identified. The papers that facilitate the analysis of accurate discussion of trends in algal bioremediation and are related to immobilization techniques were selected manually, and 139 papers that met the relevant criteria were downloaded and retained through Endnote software, and adopted in this review article.

## 2.2. Mechanisms of Heavy Metal Removal by Algae

Heavy metals are metallic elements with densities greater than 5 g/cm$^3$ and are toxic at lower concentrations [27]. Heavy metals are classified into three groups [28,29], (i) transition elements, which contain certain minor amphoteric oxides (titanium (Ti), zirconium (Zr), hafnium (Hf), rutherfordium (Rf), vanadium (V), Niobium (Nb), tantalum (Ta), chromium (Cr), molybdenum (Mo), tungsten (W), manganese (Mn), technetium (Tc), rhenium (Re), ferrum (Fe), ruthenium (Ru), osmium (Os), and zinc (Zn)); (ii) rare earth elements, including lanthanides (with lanthanum (La)) and actinides (with actinium (Ac)); and (iii) elements of the p-group dominated by gallium (Ga), indium (In), thallium (Tl), stannum (Sn), plumbum (Pb), antimony (Sb), bismuth (Bi), and polonium (Po) as the main elements of the p-group. The p-group elements are the elements of the third main group to the seventh main group plus the zero group.

Without changing their own activity, algae are capable of forming cellular protein-heavy metal complexes [30]. Organometallic complexes are further divided inside the vesicles to control the amount of heavy metal ions in the cytoplasm and lessen their hazardous effects [31]. A three-stage mechanism, involving the extracellular precipitation/accumulation of heavy metals by living cells, complexation or cellular adsorption in living and dead cells, and intracellular internalization requiring microbial activity or metabolic processes, allows algae to remove heavy metals from the environment [32,33].

Biosorption activities known as rapid extracellular passive processes can be carried out by both living and non-living biomass. The primary mechanism of heavy metal adsorption by active or passive algal biomass is biosorption, which has been demonstrated to be a practical method for removing heavy metals from industrial effluent [34]. Within a few seconds, heavy metal ions are passively absorbed after interacting with negatively charged functional groups found on the algae cell surface. Heavy metals bind to cell walls that include sulfate, carboxyl, amino, and hydroxyl groups, and the attachment of heavy metal ions occurs via chelation/complexation, adsorption, electrostatic interactions, surface precipitation, and ion exchange to functional groups on the cell surface [35]. Positively or negatively charged ions will bind to the surface of the biosorbent that has been negatively charged in the ion exchange process and has grown to be the predominant mechanism [36]. Electrostatic repulsion between positively charged surfaces and metal cations may be influenced by the protonation of the functional groups of algal biomass particles and the amino and hydroxyl groups of carriers [37]. Cd(II) is transferred from aqueous solutions to algal cell surfaces by membrane flow or boundary layer diffusion, and the immobilized algal cells have more carboxylate groups, resulting in the faster transfer of Cd(II) [38]. Additionally, biosorption can create complexes with functional groups found on the surface of cells [39]. A diverse variety of biopolymers, such as humic compounds, lipids, nucleic acids, polysaccharides, proteins, and glyoxylates, are also found in cyanobacterial extracellular polymer components in algae [40]. Cyanobacterial extracellular polymers play a crucial function in the biosorption of heavy metals and serve as a barrier against hostile external conditions [41]. Polysaccharides enable heavy metals to readily bind to algae surfaces, lipids, and proteins. Moreover, heavy metals have a tendency to precipitate and accumulate on the cell surface when the pH of the solution changes rapidly during biosorption or when the concentration of the metals rises to saturation. This process is another way that algae bind to heavy metals. The heavy metals adsorbed on the surface erode the algal cell surface, while the immobilization process results in a smoother algal cell surface, and some carriers preferentially bind to metal ions, reducing the solution metal concentration, thus making it possible to protect algal cells from adsorption [42]. Algae will produce more extracellular polymeric substances (EPS) rich in negatively charged groups in response to heavy metal ions [43]. These EPS appear to be able to generate an extracellular protective barrier on the surface of the cell wall to prevent the harmful effects of heavy metals in the intracellular environment because they feature a lot of charged hydrophobic groups that are suited for the active binding of heavy metals [44,45].

Active bioaccumulation is the transport of heavy metals across the cell membrane to the cytoplasm or other organelles and requires energy to accumulate intracellular heavy metals; however, the process is a slow intracellular active accumulation of compartmentalization [46]. Depending on the kind of biomass, chemicals are absorbed, and nutrients are taken up through the surface of the biomass, which either accumulates or metabolizes substances. Ion-selective transport proteins found in the cell membrane are necessary for the whole process, which, from the absorption of metal ions to the movement of these ions throughout the cell or any organelle, takes a long time [47].

Algae must safeguard cells against non-essential metals and maintain intracellular ion concentrations at appropriate levels. As a result of structural/binding proteins, such as metallothioneins, binding to the adsorbed ions, the host cell is spared the inhibitory effects of a high concentrations of metal ions [48]. The sulfhydryl groups in phytochelatin peptides synthesized by microalgae through enzymatic synthesis are responsible for metal binding as organometallic complexes stored in the organelles of microalgal cells [49]. Additionally, acidic calcifiers and polyps promote the accumulation and storage of heavy metals [50].

Biotransformation in algae is mainly applied to the enzymatic and biochemical transformation of heavy metals but has also been used for detoxification pathways in algae. Enzymatic biotransformation is due to the non-degradable nature of heavy metals, converting them into less harmful inorganic complex forms [51]. In contrast, biotransformation is the use of electron transfer to reduce highly valued heavy metals and which will then be converted into organic heavy metal compounds [52].

Furthermore, the mechanisms of algal adsorption can differ due to the different properties of heavy metal ions [53]. The primary mechanism of adsorption of cadmium cations by algal biomass is apparently chelation, and the adsorption of nickel ions is mainly ion exchange [54]. The binding processes of lead cations, in contrast, combine ion exchange, chelation, and reduction events with the precipitation of metallic lead on algal biomass. Lead cations have a greater affinity for algal biomass [55]. Sarojini et al. [53] verified that algae adsorb Cr ions mainly through electrostatic interactions and ion exchange. To combat arsenic toxicity, microalgae oxidize As(III) to As(V), which then undergo methylation, volatilization, and extracellular excretion as they are transformed into less toxic forms [56]. Higher Cd concentrations have a considerable impact on cellular processes linked to energy consumption, DNA replication, cell cycle, and signal transduction [57]. The process by which algal cells remove heavy metal ions is shown in Figure 1.

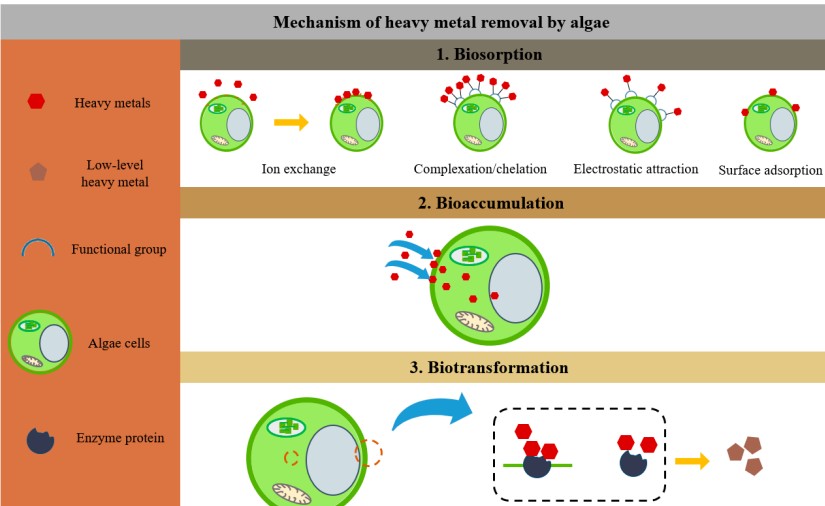

**Figure 1.** Removal of heavy metals by algae through biosorption, bioaccumulation, and biotransformation mechanisms.

## 2.3. Immobilized Algae Technology

Immobilization is carried out by attaching microalgae to the external surface of a supporting biological carrier. Cell immobilization techniques, metabolic processes with reduced susceptibility to senescence and significant stability over time, have been inspired by the attachment of living microorganisms to one another and to solid surfaces [58]. The target cells will be encapsulated by a porous polymer layer, thus allowing the process to diffuse the substrate into the cells [59]. The small particle size of the free particle biosorbent, the strong densification, and the uneven distribution on the reaction bed make the process less efficient and more difficult to separate [60]. The combined synergistic impact of immobilized systems can improve resistance to cell growth disruption, prevent photoinhibition and minimize cytotoxicity, and considerably aid microalgal cells in tolerating and adapting to environmental stress or toxicity [61,62]. Immobilized algae boost volumetric output, increase substrate usage, and increase resistance to harmful elements (e.g., extreme pH, temperature, and toxic compounds) [63]. Immobilized *Sargassum* contrasts with free *Sargassum* for Cu(II) ions, and immobilized adsorbents have high metal uptake, improving biosorption of nickel ions by 49% and copper ions by 36% [64].

Furthermore, during immobilization, the mobility of algal cells is affected by the limited intracapsular space, which can lead to high shear stresses from chemical forces and interactions between the support matrix and microalgal cell walls [65]. Immobilization processes prevent biomass loss from the process and improve operational flexibility, and the immobilization or sequestration of cells in small confined spaces may trigger interactions that enhance nutrient uptake [66]. Therefore, cell immobilization technology will accelerate the rate of nutrient uptake by microalgae, thus increasing the efficiency of wastewater treatment systems, which can further increase productivity and thus reduce production costs [67,68]. Additionally, compared to suspended systems, the substrate may restrict or lessen the degree of photon accessibility of algal cells, which will result in less biomass formation. However, the morphological and physicochemical characteristics of immobilized algae can be altered by homogenizing the intracapsular and extracapsular phases as well as by enhancing the substrate's characteristics in order to improve intracapsular mobility and achieve effective mass transfer performance. The creation of improved reactor designs, as well as the provision of infrastructure and logistics, are necessary for the scale-up of immobilized algae technology to create algal beads on a commercial scale [69]. Algal immobilization techniques include adsorption, encapsulation, entrapment, and self-immobilization [70]. Figure 2 summarizes the advantages and disadvantages of immobilization techniques.

### 2.3.1. Adsorption

Adsorption is a process that forms a physical bond between the surface of the water-insoluble carrier and immobilized algae through weak molecular forces like van der Waals interactions and ionic and hydrogen bonding, which are relatively gentle and quick. As a result, during use, there is a significant amount of cell leakage from the carrier due to the adsorption process [71]. Shen et al. [72] uncovered that effective adsorption of $Fe_2O_3$ on microalgal surfaces resulted in nanoscale spherical iron oxide covering the microalgae, opening the door to the potential of immobilizing microalgae using metal oxides. Through adsorption on the surface of the substance and passage through the algal cells, surface-immobilized algae lowered the heavy metal burden in the effluent. The growth of algal cells adsorbed onto the biofilm surface reduces the recovery cost because the method is easier to perform [73]. Adsorption-type immobilized algae have a lower cell concentration than encapsulated cells, and cells leak from the surface of the carrier during algal growth [74].

### 2.3.2. Encapsulation

Encapsulation is a permanent kind of immobilization in which cells are confined in a capsule space created by membrane walls. The cells can float freely in the inner space of the capsule despite being physically constrained [75]. Whitton et al. [76] investigated how

light affected calcium alginate beads, encapsulating immobilized microalgae for nutrient remediation. The use of the enclosed carriers increased the substrate conversion and simplicity of collection by shielding the microbes from environmental stress/shock loading and hazardous byproducts. Alginate bead encapsulation techniques have drawbacks, such as poor swelling and mechanical qualities, which can cause damage or mass loss during adsorption [77]. Additionally, the encapsulation method limits the mass transfer rate, is unstable at a specific pH, and easily dissolves in buffers. Qin et al. [78] developed novel algae-encapsulated macro-capsules combined with membrane separation, where dual encapsulation created a restricted microaerobic environment with higher biomass harvesting and activity, the improved stability of live cells, and reduced cell leakage rates.

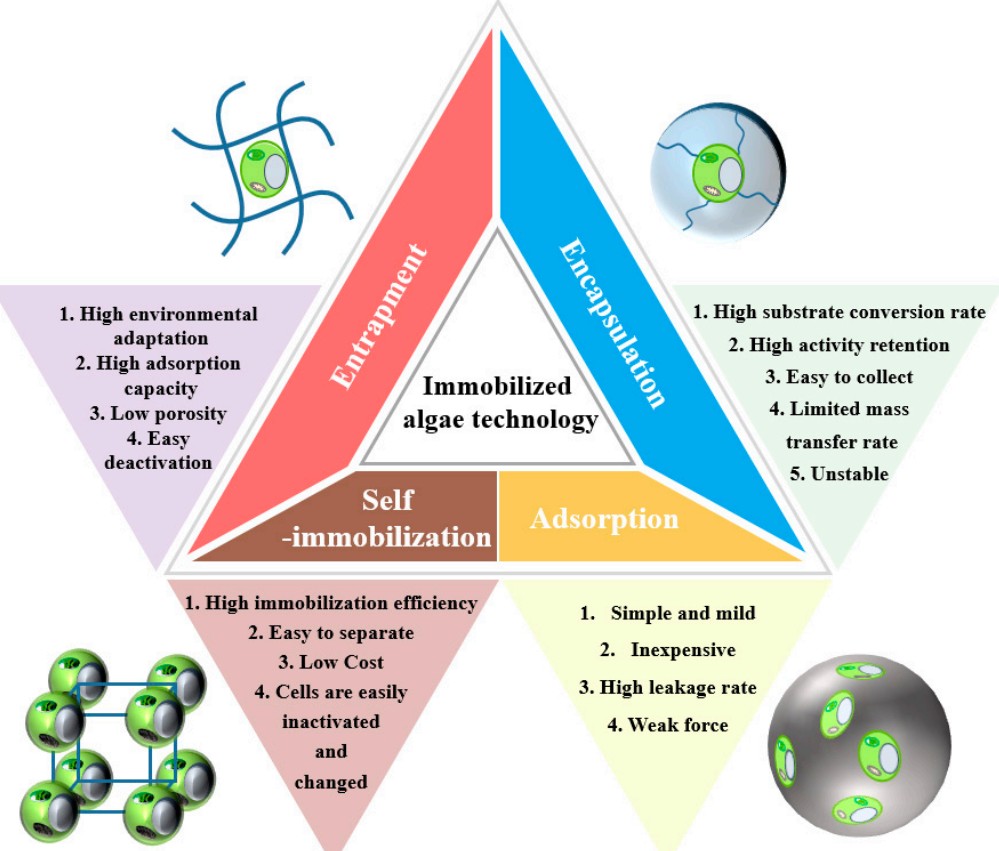

**Figure 2.** Advantages, disadvantages, and characterization of methods for the immobilization of algae by entrapment, encapsulation, adsorption, and self-immobilization.

### 2.3.3. Entrapment

The method of the entrapment of cells in a polymer matrix and self-adhesion of the cells to the surface of solid support is entrapment and is the most commonly used immobilization method [79]. This method captures algal cells into a supporting matrix, namely a fiber or natural gel polymer. Maswanna et al. [80] entrapped green alga *Tetraspora* sp. CU2551 in alginate substrates with 10–50 times higher hydrogen production than cyanobacteria, which was considered a promising biological system. It has a bigger specific surface area, can adsorb a higher density of bacteria and algae, can sustain a greater pollution load, and is more adaptive to environmental conditions than adsorption immobilization and self-immobilization on the carrier surface [81]. Kube et al. [82] discovered that immobilizing algal cells by enclosing them in alginate beads assisted in the beginning and sustained larger densities of algae in the reactor, which enabled the quick removal of heavy metals. Saxena et al. [83] used freshwater diatom *Nitzschia palea* entrapped in calcium alginate hydrogel beads by gelation method without swelling behavior and in a more stable form

to consume the nitrate, phosphate, and ammonia load in the water column. The gelation reaction was also shown to be reversible. Entrapment immobilization suffers from the high inactivation of algal cells [84]. The low porosity of immobilized algal cells via natural polymers can lead to restricted nutrient diffusion and thus affect the bio-removal efficiency of immobilized cells [74].

### 2.3.4. Self-Immobilization

The filamentous fungus can serve as immobilization carriers for mycorrhizal self-immobilization since they can spontaneously cluster into spheres and immobilize various mycorrhizal species [85]. Applying multifunctional reagents and crosslinking immobilization encourages the creation of channels between functional groups on the outer cell membrane [86]. The successful use of crosslinked polyethyleneimine polymer on immobilize *C. vulgaris* cells was achieved [87]. Electrostatic interaction between the negatively charged microalgae surface and the positively charged adsorbent amine results in a significant improvement in immobilization efficiency [88]. Carrier-free engagement can reduce the cost of materials and replace time-consuming and expensive technologies [89,90]. Artificially induced conditions of leading to the formation of algal cell aggregates have fewer mass transfer limitations and can better enhance cell growth, resulting in higher cell density. However, the cellular makeup of microalgae can be unintentionally altered when algal cells are exposed to chemicals and severe environments that could harm the cell surface and decrease their metabolic activity [91].

### 3. Immobilization Parameters

The biological removal process of heavy metals by immobilized algae is influenced by many factors, such as adsorbent content, initial heavy metal ion concentration and type, temperature, pH, contact time, metal system, and algae and carrier type. The effect of various factors on the efficiency of heavy metal adsorption by immobilized algae are shown in Table 1.



**Table 1.** The influence of important parameters of heavy metal removal by immobilized algae on adsorption efficiency.

| Algae Type | Immobilized Carriers | Heavy Metal Types and Systems | Immobilization Method | Adsorbent Dosage (g/L) | Optimal Initial Metal Concentration (mg/L) | Optimal pH | Optimal Temperature (°C) | Optimal Contact Time | Maximum Adsorption Capacity (mg/g) | Maximum Adsorption Efficiency | References |
|---|---|---|---|---|---|---|---|---|---|---|---|
| *Chlorella sorokiniana* | Alginate | Cu(II) | Encapsulation | - | 25 | 5.0 | 40 | 180 min | 150.07 | 97.10% | [92] |
| | | Cd(II) | | | 25 | 5.0 | 20 | 180 min | 48.87 | 50.94% | |
| | | Ni(II) | | | 25 | 5 | 20 | 180 min | 101.73 | 74% | |
| | | Cu(II)/Ni(II) | | | 30 | 5.0 | 40 | 180 min | Cu(II):21.47 Ni(II):11.15 | Cu(II):89.68% Ni(II):39.66% | |
| | | Cu(II)/Cd(II) | | | 50 | 5.0 | 20 | 180 min | Cu(II):39.13 Cd(II):15.11 | Cu(II):91.53% Cd(II):32.64% | |
| | | Cd(II)/Ni(II) | | | 30 | 5.0 | 20 | 180 min | Cd(II):15.10 Ni(II):11.77 | Cd(II):63.03% Ni(II):42.08% | |
| | | Cu(II)/Ni(II)/Cd(II) | | | 30 | 5.0 | 40 | 180 min | Cu(II):24.30 Ni(II):12.59 Cd(II):8.25 | Cu(II):84.51% Ni(II):47.41% Cd(II):32.43% | |
| *Chlorella vulgaris* | Calcium alginate beads | Fe(II) | Adsorption | 0.6 | 250 | 6 | 25 | 450 min | 43.43 | - | [21] |
| | | Mn(II) | | 0.6 | 250 | 6 | 25 | 450 min | 40.98 | - | |
| | | Zn(II) | | 0.6 | 250 | 6 | 25 | 450 min | 37.43 | - | |
| *Micractinium reisseri* KGE33 | Silicon dioxide | Cu(II) | Entrapment | 100 | - | 5 | 40 | 24 h | 1.710 | 87.1% | [93] |
| *Synechocystis* sp. PCC6803 | Fe$_2$O$_3$ | Cr(VI) | Adsorption | 0.5 | 100 | 2.0 | 29.85 | 30 min | 69.77 | 88.37% | [72] |
| | | Cu(II) | | 0.5 | 100 | 5.0 | 29.85 | 60 min | 38.68 | 78.89% | |
| | | Pb(II) | | 0.5 | 100 | 5.0 | 29.85 | 30 min | 62.63 | 88.89% | |
| | | Cd(II) | | 0.5 | 100 | 5.0 | 29.85 | 30 min | 42.12 | 88.89% | |
| *Chlorella* sp. (FACHB-31) | Biochar | Cd(II) | Surface adsorption, polymer matrix | 1.0 | 100 | 6.0 | 26 | 50 min | 217.41 | 86.57 ± 0.61% | [42] |
| *Cladophora* sp. | Chitosan | Cd(II) | Crosslinking | 0.2 g | 10 | 6.0 | 25 | 60 min | 0.240 mmol/g | - | [37] |
| | | Cr(III) | | 0.2 g | 10 | 5.0 | 25 | 360 min | 1.128 mmol/g | - | |
| | | Cu(II) | | 0.2 g | 10 | 5.0 | 25 | 360 min | 1.059 mmol/g | - | |
| | | Ni(II) | | 0.2 g | 10 | 6.0 | 25 | 60 min | 0.239 mmol/g | - | |
| | | Zn(II) | | 0.2 g | 10 | 5.0 | 25 | 60 min | 0.310 mmol/g | - | |
| *Chlorella* sp., *Ankistrodesmus braunii*, and *Scenedesmus quadricauda* var *quadri-spina* | Sodium alginate | Cu(II) | Entrapment | 10 g | 50 | 3.0 | 28 ± 2 | 180 min | - | 43.19% | [94] |
| *Chlorella* sp. (FACHB-31) | Water-hyacinth leaf pelle | Cd(II) | Surface adsorption | 1.3 | 10 | 6.0 | - | 5 d | - | 48% | [95] |
| | Water-hyacinth root pellet | | | 1.3 | 10 | 6.0 | - | 5 d | - | 35% | |
| | Water-hyacinth leaf biochar pellets | | | 1.3 | 10 | 6.0 | - | 5 d | 13.81 ± 0.94 | 92.45 ± 0.5% | |
| | Water-hyacinth root biochar pellets | | | 1.3 | 10 | 6.0 | - | 5 d | - | 60% | |

**Table 1.** *Cont.*

| Algae Type | Immobilized Carriers | Heavy Metal Types and Systems | Immobilization Method | Adsorbent Dosage (g/L) | Optimal Initial Metal Concentration (mg/L) | Optimal pH | Optimal Temperature (°C) | Optimal Contact Time | Maximum Adsorption Capacity (mg/g) | Maximum Adsorption Efficiency | References |
|---|---|---|---|---|---|---|---|---|---|---|---|
| *Anabaena variabilis* | | Fe(II) | | | 13.88 | | | 6 h | | 94.45% | |
| *Anabaena variabilis* | | Zn(II) | | | 5.1 | | | 6 h | | 98.98% | |
| *Anabaena variabilis* and *Tolypthrix ceytonica* | | Zn(II) | | | 5.1 | | | 6 h | | 98.63% | |
| *Tolypthrix ceytonica* | Water-hyacinth leaf pelle | Zn(II) | Entrapment | - | 5.1 | - | - | 6 h | - | 98.61% | [96] |
| *Anabaena variabilis* and *Tolypthrix ceytonica* | | Pb(II) | | | 4.5 | | | 6 h | | 94.22% | |
| *Anabaena variabilis* | | Cu(II) | | | 0.15 | | | 6 h | | 93.33% | |
| *Tolypthrix ceytonica* | | Cu(II) | | | 0.15 | | | 6 h | | 91.33% | |
| *Chlorella sorokiniana* and *Monoraphidium* sp. | Sodium alginate beads | Cu(II) | Entrapment | 0.5 g | 20 | 4.0 | 35 | 180 min | - | 96.4% | [97] |
| *Sargassum* sp. | Calcium alginate beads | Ni(II) | Entrapment | 0.1 g | 50 | 5.0 | 30 | 4 h | 1.69 mmol/g | - | [64] |
| | | Cu(II) | Entrapment | 0.1 g | 50 | 5.0 | 30 | 6 h | 2.06 mmol/g | - | |
| *Penium margaritaceum* | Filter paper | Pb(II) | Adsorption | 1.0 | 1.0 | - | 25 | 8 h | 3.4 | 55.4% | [98] |
| *Chlorella vulgaris* | Calcium alginate beads | Cd(II) | Entrapment | 0.5 | 75 | 6.0 | 25 | 105 min | 1.168 | 76.448% | [99] |
| *Chlamydomonas reinhardtii* | Carboxymethyl cellulose beads | U(VI) | Entrapment | - | 1 | 4.5 | 25 | 60 min | 218.3 | 92.4% | [100] |
| *Sargassum* sp. | Sodium alginate | Ni(II) | Entrapment | - | 1 mmol/L | 4.5 | 30 | - | 1.404 mmol/L | - | [101] |
| | | Cu(II) | Entrapment | - | 1 mmol/L | 4.5 | 30 | - | 1.656 mmol/L | - | |
| *Spirulina platensis* | Beads | Cr(VI) | Entrapment | 1.0 | 250 | 3.0 | 25 | - | 49 | 75% | [102] |
| *Turbinaria ornata* | Sodium alginate beads | Cd(II) | Entrapment | 5.04 | 25.2 | 5.06 | 25 | 90 min | - | 98.65% | [38] |
| *Synechocystis* sp. PCC6803 | Sodium alginate | Cr(VI) | Adsorption-crosslinking | 1.5 | 40 | 7.0 | 30 | 30 min | 7.6 | - | [103] |
| | Chitosan | Cr(VI) | Adsorption | 1.5 | 40 | 7.0 | 30 | 30 min | 37.1 | - | |
| | | Cu(II) | | 1.5 | 40 | 7.0 | 30 | 30 min | 25.98 | - | |
| | | Pb(II) | | 1.5 | 40 | 7.0 | 30 | 30 min | 25.06 | - | |
| | | Cd(II) | | 1.5 | 40 | 7.0 | 30 | 30 min | 24.62 | - | |
| | Carrageenan | Cr(VI) | Polymer matrix | 1.5 | 40 | 7.0 | 30 | 30 min | 19.7 | - | |
| | Diatomite | Cr(VI) | Adsorption | 1.5 | 40 | 7.0 | 30 | 30 min | 8.0 | - | |
| | Quartz sand | Cr(VI) | Entrapment | 1.5 | 40 | 7.0 | 30 | 30 min | 6.2 | - | |
| | Polyvinyl alcohol | Cr(VI) | - | 1.5 | 40 | 7.0 | 30 | 30 min | 24.2 | - | |

Table 1. *Cont.*

| Algae Type | Immobilized Carriers | Heavy Metal Types and Systems | Immobilization Method | Adsorbent Dosage (g/L) | Optimal Initial Metal Concentration (mg/L) | Optimal pH | Optimal Temperature (°C) | Optimal Contact Time | Maximum Adsorption Capacity (mg/g) | Maximum Adsorption Efficiency | References |
|---|---|---|---|---|---|---|---|---|---|---|---|
| *Sargassum vulgare* | Calcium alginate beads | Fe(III) | Entrapment | 20 | 50 | 2.0 | 25 | 120 min | 17.09 | 86.07% | [104] |
| *Spirulina* | Calcium alginate beads | Pb(II) | Entrapment | 10 | 5.63 | 5.2 | 25 | 72 h | 282.17 | - | [105] |
| *Cladophora* sp. alga | Calcium alginate beads | Hg(II) | Entrapment | 10 | 100 | 5.0 | 16 | 60 min | 43.87 | - | [106] |
| | Silicone | Hg(II) | Entrapment | 10 | 100 | 5.0 | 16 | 60 min | 39.47 | - | |
| *Sargassum filipendula* | Sodium alginate | Cu(II) | Entrapment | 0.1 g | - | 5.0 | 30 | - | 3.60 mmol/g | - | [107] |
| | | Ag(I) | | 0.1 g | - | 5.0 | 30 | - | 8.67 mmol/g | - | |
| *Chlorella sorokiniana* | Sulfur-Sigma-Aldrich's castor oil copolymer | Cd(II) | Adsorption | 1 | 50 | 6.0 | 27 | 24 h | - | 80% | [108] |
| | Sulfur- Castor oil copolymer | | | 1 | 50 | 6.0 | 27 | 24 h | - | 90% | |
| | Sulfur and Sigma-Aldrich's castor oil copolymer | Cu(II); Cd(II) | | 1 | 8 | 6.0 | 27 | 24 h | - | Cu(II):92%; Cd(II):90% | |
| | Sulfur- Castor oil copolymer | | | 1 | 8 | 6.0 | 27 | 24 h | - | Cu(II):95%; Cd(II):90% | |

### 3.1. Adsorbent Dosage

The primary determinant of the biosorption effectiveness is the amount of sorbent present. Higher dosages might lead to the production of cellular aggregates, which reduces the effective surface area for biosorption and explains this tendency [21,109]. The quantity of functional groups on the adsorbent's surface during the adsorption process is determined by the amount of adsorbent [110]. Due to an increase in unsaturated active sites on the biosorbent, a rise in algal dosage suggests a decrease in each metal ion's ability for biosorption [111]. The amount of adsorbent is due to the increase in the surface area of the adsorbent, which increases the available functional sites' mass of the adsorbed metal and improves the adsorption efficiency. The specific adsorption of Cd(II) is improved when the amount of biosorbent is increased, but an excess of biosorbent is typically accompanied by limited availability, electrostatic interactions, interference between binding sites, and reduced mixing, which causes the specific metal adsorption to be reduced [112]. The amount of 0.2 g of chitosan–algae composite microbead adsorbent is close to saturation for heavy metal adsorption [37]. A 10-fold increase in the number of carboxymethyl cellulose beads immobilized algae decreased the amount of adsorbed U(VI) ions from 218.3 to 93.1 mg/g, despite greater surface area and the availability of additional adsorption sites [100]. The effectiveness of heavy metal removal can be significantly increased by increasing the amount of adsorbent, but doing so has no discernible impact because all the heavy metal ions in the solution react with the adsorbent's active sites, and the amount of metals in the solution is insufficient to cover these sites [103]. However, it is also possible that the adsorption capacity is reduced due to the concentration gradient between the biosorbent and the heavy metal ions.

### 3.2. Initial Metal Concentration and Type

The ionic radius of Cd(II) (0.95 Å) is larger than that of Cu(II) ions (0.73 Å) and Ni(II) ions (0.69 Å) and therefore has a higher physical affinity for Cd(II) ions at the biosorption sites of the cells [92]. The biosorption process intensifies in the early stages, and metal biosorption becomes insignificant as the initial metal ion concentration rises further. In order to increase the metal absorption, the mass transfer barrier between the biosorbent and the aqueous solution must be overcome by the metal ions, which are driven by the greater starting concentration. The removal efficiency of $Fe_2O_3$@microalgae for heavy metals above 100 mg/L did not change much because there were not enough active sites in the biosorbent to accommodate the increase in the number of ions for diffusion or their collision with one another [113]. Sargın et al. [37] were able to adsorb Cd(II) and Zn(II) ions with intact full *d* subshells through ionic bonding, while the incorporation of algal biomass was still not able to help the enhancement of the adsorption of Ni(II) with lower ionic radii. Carboxymethylcellulose beads immobilize algal active binding sites to saturation at 1000 mg/L concentration [100]. While most high concentrations of heavy metal ions are highly toxic and will sharply reduce the impact on algal growth, some low concentrations of metal ions will stimulate the growth of algal cells. Meanwhile, immobilized algae will reduce the activity of heavy metals due to the metal chelating and ion exchange activity of the immobilized carrier [58].

### 3.3. Temperature

Temperature significantly affected the metabolism and cell death of microorganisms; the adsorption of Fe(II), Mn(II), and Zn(II) showed the same trend regarding temperature; and the enzymes present in the cells for transferring ions showed maximum enzymatic activity at an ambient temperature; therefore, it can be stated that increasing the temperature decreased the efficiency of adsorption of heavy metal ions [21]. The interaction between microalgae and iron oxide showed a good adsorption effect at 29.85 °C [72]. The surface activity and kinetic energy of the adsorbate are often increased by raising the temperature, which improves biosorption. However, this may also cause the physical structure of the biosorbent to be disturbed. Increasing temperature increases the active sites and enhances

the activity of the biosorbent, further enlarging the biosorbent pores and reducing the thickness of the diffusion boundary layer around the biosorbent, in addition to bringing about an increase in the mobility of metal ions and an increase in the surface activity of the biosorbent. Lieswito et al. [97] explored the removal of Cu(II) via immobilized algae in the range of 25–45 °C and found the highest removal rate (94.8%) at 35 °C. Typically, the biosorption of metals by algae is inherently heat-absorbing, and the biosorption efficiency increases with increasing temperature [114]; however, the optimal temperature for the bioreaction of immobilized algae is restricted to a limited range [38]. Therefore, the adsorption of algae is suitable for favorable adsorption at moderate temperatures, Wang et al. [103] studied the adsorption of Cr(VI) on chitosan at temperatures higher than 30°C with a gradual decrease in adsorption, demonstrating that temperature affects the Brownian motion of molecules and that high temperatures lead to the increased thermal motion of molecules, thus disrupting the adsorption equilibrium and reducing the adsorption capacity.

### 3.4. pH Value

The pH variation significantly impacts the metal ion shape and the protonation and ionization of functional groups on the adsorbate. The significance of the impact of pH on the biosorption of heavy metals was highlighted by Petrovič et al. [92]. This impacts the state of metal ions in the solution and the characteristics of the solution, in addition to impacting the surface charge distribution of the adsorbent [115]. The ideal pH range for metal biosorption is thought to be the choice of the zero charge point. The dissociation of functional groups from the biosorbent surface will increase as pH rises. While pH 5.5 exhibited a precipitation impact on Cu ions, pH 5 was considered the optimal pH for eliminating Cu ions. The pH impacted the chemistry of metals in solution and the metal binding on the surface sites of algal cells. A maximum of 44.43 mg/g Fe(II), 40.98 mg/g Mn(II), and 37.43 mg/g Zn(II) were adsorbed by immobilized live algal cells at pH 6 [21]. According to Shen et al. [42], raising the pH allows the positive charge on the surface to neutralize the negative charge, lowering the surface potential, reducing acceleration, and thereby reducing the absorption of Cd (II). Therefore, immobilized algae exhibit lower adsorption efficiency than normal free algae due to the low pH's positive surface charge, which limits Cd's binding (II). Repulsive forces prevent metal ions from approaching at lower pH levels, and a high number of H ions compete for adsorption sites, limiting the metal adsorption, since at lower pH levels, cell wall ligands are tightly connected with hydrated hydrogen ions. The protonation of functional groups of algal biomass particles and the amino and hydroxyl groups of chitosan have been shown by Sargın et al. [37] to potentially contribute to the electrostatic repulsion between positively charged surfaces and metal cations. The impact of $H_3O^+$ is reduced when pH rises because additional ligands—such as amino and carboxyl groups—are accessible and consequently negatively charged. According to Lee et al. [93], the electrostatic gravitational attraction between green algae and copper ions at higher pH levels might be partially responsible for the adsorption of Cu(II) by silica immobilized microalgae. Cr(VI) ions prefer the surface complexes of positively charged $Fe_2O_3$@microalgae at low pH, whereas Cu(II), Pb(II), and Cd(II) bind more readily to the adsorbent binding sites at pH > 4 [72]. The electrostatic interaction between the negatively charged algal cells and carboxymethyl cellulose bead polymers and the positively charged uranyl ions changes with pH, and the ideal complexation pH is 4.5 [100]. Cr(VI) formation in different chromate solutions is highly dependent on the pH of the solution, and since the isoelectric point of microalgal proteins is near pH 3, immobilized algae show a negative charge due to the electrostatic repulsion in a strongly acidic environment, greatly reducing the biosorption capacity [102]. Low pH levels below the anticipated point of zero charge cause the biosorbent's surface to become totally protonated, which lowers the adsorption capacity of immobilized algal biomass [38]. Contrarily, protons must contend with $OH^-$ for suitable adsorption sites in intervening gaps or on the adsorbent surface under alkaline pH conditions, which also prevents the adsorption of Cr(VI) [103].

### 3.5. Contact Time

The kinetic rate of biosorption depends on the determination of the contact time. The biosorbent becomes saturated with the biosorbate as the biosorption process moves forward, and the desorption process then tends to occur. The biosorption and desorption rates will be equal at the equilibrium point. Immobilized algae kinetics are slower than free algae, so free algae have more free binding sites to adsorb heavy metal ions during pre-exposure [116]. The biosorbent will no longer bind to the biomass after the biosorption process achieves an equilibrium condition. Due to increased repulsive interactions between the adsorbent and the adsorbing ions, the adsorption rate reduces over time as the number of vacant sites decreases [117]. The greatest contact duration to achieve the highest adsorption effectiveness is 180 min. The adsorption capacity of alginate-immobilized algae for heavy metal ions rises with increasing contact time [92]. The calcium alginate immobilization of live algal cells requires 450 min to reach equilibrium [21]. The reactive groups determine the rate of heavy metal adsorption on the surface of the $Fe_2O_3$@Microalgae composite, and the optimum binding equilibrium can be achieved for Cr(VI), Pb(II), and Cd(II) in 30 min [72]. In contrast, Lieswito et al. [97] studied the removal of Cu(II) using immobilized algae at optimal temperature and found that 96.4% of Cu(II) ions could be removed during a contact time of 180 min. As empty surface binding sites are in contact with Cd(II) ions, Cd(II) biosorption is initially quicker, with rapid removal occurring within the first 15 min and then gradually improving until equilibrium is reached at 90 min [38]. All of the monolayer's active spots biosorb, lengthening contact time and decreasing removal effectiveness [118]. The slower kinetics of immobilized biomass compared to pristine algae can be explained by biomass binding within the immobilized matrix, whereas the binding sites of pristine biomass are open to $Fe^{3+}$ ions. This was the conclusion reached by Benaisa et al. [104] in their study on the impact of contact time on immobilized algae. Immobilized algal cells can be tested faster, thus reducing the contact time of the solution sorbent.

### 3.6. Metal Systems

Although the sorption capacity of polymetallic systems is generally considered to be better than the larger monometallic systems [119], Petrovič et al. [92] found that increasing the number of co-metals in solution reduced the biosorption capacity of the metals involved, so that polymetallic systems were more complex than monometallic adsorption systems and that antagonism of metals inhibited the adsorption of other metals, thus reducing the removal efficiency. Tofan [120] also discovered that the rivalry between several metal ions for the active site on the biosorbent led to the maximum adsorption of Co(II) for polymetallic solutions resisting that of the equivalent monometallic solution. The maximum biosorption of algae immobilized in alginate beads in polymetallic solutions reduced from 9.812 mg/g to 6.855 mg/g, as shown by Mokone et al. [106], whereas the adsorption efficacy of algae immobilized in silica gel decreased even more by 40%. The presence of competing ions influences the choice of target heavy metal ions by immobilized algae; heavier metal ions with a higher affinity will preferentially adsorb to binding sites, restricting the removal of the target ions. However, the adsorption efficiency of a mixed bimetallic system of copper and cadmium removal by algal surface adsorption on sulphur copolymers was not significantly different from that of the cadmium adsorption alone [108]. The adsorption performance of Cu(II)/Ag(I) polymetallic adsorption systems was compared to that of single metal systems by Do Nascimento et al. [107]. Cu(II) ions may cause other oxygen and sulfonate groups to bind to silver ions, thus improving their biosorption capacity, and so the bimetals encourage each other's adsorption when they are coupled to carboxyl groups to form a water-insoluble copper alginate network.

### 3.7. Algae Type

Algae are a natural biomass and an important sorbent material. The toxicity levels of heavy metal ions in different algae may be highly strain-specific, with different affinities for a wide range of metals, thus determining the potential remediation capacity of using

specific algal strains. *C. vulgaris* has better biosorption efficiency than *A. platensis* [21]. Cyanobacteria are very promising heavy metal-absorbing microorganisms in algae and are capable of oxygenated photosynthesis. *A. variabilis* had a higher sorption capacity for heavy metals than *T. ceytonica* and the combination of the two cyanobacteria [96]. Brown algae have always been of interest in sorbents because they contain polysaccharide alginate and fucoidan, which are active in ion exchange processes [64]. The extent of heavy metal uptake varies between algae, and Suresh Kumar et al. [121] showed differences in the uptake of heavy metals by a variety of algae.

### 3.8. Immobilized Carriers

The way that different carriers affect the immobilization of algae varies. The sustainability of the procedure would be significantly improved by using fresh materials made from industrial or agricultural waste that may be formed into adsorbent materials to immobilize microalgae [122]. Sodium alginate is often generated commercially and is a naturally occurring polymer that is derived from the cell walls of coastal brown algae [123]. It has good biocompatibility, low cost, significant binding capacity, and good hydrophilicity [124]. It possesses the considerable binding capability, cheap cost, strong hydrophilicity, and good biocompatibility [92]. Shen et al. [95] compared the immobilization of microalgae on four carriers with different surface hydrophilic properties and found that water hyacinth leaf biochar particle carriers had the highest surface hydrophilicity (91.73 $\pm$ 2.63%) and showed higher immobilization efficiency (89.30 $\pm$ 6.50%), also demonstrating that biochar-based materials provided a more compatible surface to immobilize microorganisms.

Additionally, the rapid passive biochar adsorption may lower the Cd(II) concentration, increasing the permeability of the cell wall and reducing damage to the activity of microalgae attached to the surface of the carrier [42]. Immobilization efficiency is inversely correlated with surface hydrophilicity, and biochar surfaces become more hydrophilic than pristine biomass when aromatic nuclei are exposed and aliphatic functional groups are removed. As a result, the carriers of biochar pellets are made from water hyacinth leaves, which are more hydrophilic (91.73%) and have higher immobilization efficiencies (89.3%) [95]. Biochar's porous design and strong dispersibility made immobilized algae more widely dispersed, which encouraged biosorption [22]. However, due to its high cost, it is inappropriate for use as a microalgae biomembrane support material [125]. Due to its excellent physical and mechanical stability, absence of swelling, good resistance to organic matter, and resilience to strong acids and high temperatures, silicon dioxide is regarded as a common carrier [93]. As low-cost metal oxides with high specific surface area and multiple adsorption groups, iron oxides were verified by Shen et al. [72] to have high heavy metal adsorption properties in interaction with microalgae. Chitosan is a deacetylated form of chitin produced by the chemical and enzymatic deacetylation of chitin [126]. Wang et al. [103] assessed the adsorption performance of different carriers to immobilize algae. Chitosan immobilized microalgae using carriers with chemical bonds, which was able to improve the stability of the carriers. However, the immobilization of microalgae by diatomaceous earth and quartz sand through weak interactions, such as hydrogen bonding and van der Waals forces, led to the detachment of microalgae during the cleaning process after immobilization. Chitosan readily interacts with crosslinking agents such as glutaraldehyde to form highly porous hydrophilic polymers that significantly increase mechanical strength and are used to improve the stability of new immobilized microalgae [74]. Furthermore, sodium alginate, carrageenan, and polyvinyl alcohol failed to significantly enhance the heavy metal adsorption capacity, despite their high immobilization efficiency on algae. Sulphur-castor oil copolymer immobilized algae were more effective at early stages of $Cd^{2+}$ adsorption, completely absorbing a large number of the most toxic heavy metals, demonstrating that the combination of copolymer and microalgae techniques can improve the efficiency of both remediation techniques alone, while reducing process costs [74,108].

## 4. Life Cycle Assessment and Economic Evaluation

Life cycle assessment is an ordered tool used to examine and calculate the impacts and effects caused by any product, process, or activity throughout its life cycle from extraction to utilization and reuse to environmental sink [127]. The life cycle assessment of immobilized algae can provide a quantitative measure of their sustainability. The system boundary of heavy metal adsorption by immobilized algae includes:

1. the production of algae cultivation and the production of immobilized carriers,
2. the production of immobilized algal systems and their transport to wastewater treatment,
3. the production of various solvents included,
4. the production of electricity and water,
5. the adsorption of heavy metal ions, and
6. the regeneration and reuse process of immobilized adsorbents.

The pre-production chain of algae has a significant economic impact, and open or closed systems based on suspended growth determine the biomass growth rate of immobilized algae. The growth of algae is controlled by light, water, and nutrients. The successful immobilization of cultivated algae allows for the uniform distribution of nutrients and light and the control of the growth cycle of the algae for cultivation, which allows for the better control of the life cycle than free-growing algae [128]. The immobilization process results in a longer system life cycle and an increase in cyclic sorption [129]. Thus, the immobilization process ensures a high growth rate of algae on or within the substrate as well as low cell leakage [130]. Algal biomembrane photobioreactors are thought to significantly reduce the water and energy requirements of algal culture processes, requiring 45% less water and 99.7% less energy for dewatering than open ponds to cultivate one kilogram of algal biomass [131]. Abinandan et al. [132] demonstrated through life cycle assessment that immobilized acid-adapted microalgae technology has a lower global warming potential and is more environmentally sustainable than conventional eggshell-microalgae treatment technology by treating mine drainage with an acid-adapted algal immobilization system. Additionally, entrapping microalgae cells in alginate beads can reduce fossil energy consumption by up to 50%, achieving economic viability. Furthermore, the life cycle assessment of the stationary system from raw material extraction, transport, and final product disposal stages was continued. The higher renewable energy contribution of fixed algae systems compared to active and passive treatment systems, with an increase of 9% and 80%, respectively, is a more sustainable and greener form and reduces the life cycle carbon emissions of raw materials (by 80% and 5%), due to the treatment of construction materials and the reduced consumption of diesel fuel. According to Fawzy et al. [38], immobilization can also reduce the cost of biomass removal from the treated solution by up to 60%. The regeneration of the biosorbent is crucial in large-scale applications to lower total process costs and reliance on a steady supply of raw materials. The water hyacinth leaf biochar pellet immobilized algae adsorbent can be regenerated and recycled over three cycles and still maintain a 91.1% adsorption efficiency for Cd(II) (only a 0.6% reduction) [95]. Ahmad et al. [114] performed desorption regeneration experiments on immobilized algae adsorbents. The removal efficiency of Fe(II), Mn(II), and Zn(II) was lowered by just 3.56–4.87% after five adsorption/desorption cycles, revealing the strong reusability of immobilized algae and significantly lowering the waste of adsorbent materials. The circular economy principle imposes strict limits on the discharge and reuse of pollutants from wastewater, thus putting pressure on conventional wastewater treatment systems [133]. The immobilized algae technique reduces harvesting costs and eliminates the chance of microbial contamination. Murujew et al. [69] discovered that alginate, using a stationary algal reactor, could provide a 60% reduction in net operating costs, and the minimum amount of sodium alginate carrier replenishment would be economically beneficial. The recovery of heavy metals and biosorbents after adsorption makes the process more efficient than other removal technologies, and metals recovered in elemental form can be reused in other processes [134]. However, the immobilization process increases the cost of algal wastewater treatment due to the increased use of immobilization carriers, as the cost of raw material

for the carriers and the heavy metal uptake efficiency affects the size of the immobilization system and the rate of required carrier concentration, and the degree of contamination of the carriers determines how often the immobilized algae need to be replenished [135]. The removal of heavy metals by newly immobilized algae is illustrated in Table 2 by a life cycle assessment and cost analysis. The goal of immobilized algae technologies for the development of net zero carbon emissions and circular economy need to shift the focus of testing protocols and standard assessment frameworks to a more comprehensive life cycle environmental assessment.

**Table 2.** Validation of life cycle assessment and economic analysis for the sustainability of heavy metal removal by immobilized algae.

| Algae Types | Immobilization Carriers | Adsorption of Heavy Metals | Comparison Method | Life Cycle Assessment | Economic Analysis | Reference |
|---|---|---|---|---|---|---|
| *Desmodesmus* sp. MAS and *Heterochlorella* sp. MAS3 | Alginate beads | Fe(II) | Eggshell-microalgae method | GWP: three-fold reduction; 51.53 kg/m$^3$ $CO_2$ reduction from transport; 3.397 kg/m$^3$ $CO_2$ reduction from coal-fired power generation | 50% reduction in fossil fuel consumption | [132] |
| | | | Limestone systems | GWP: seven-fold reduction, 3.207 kg/m$^3$ $CO_2$ reduction from coal-fired power generation | 50% reduction in fossil fuel consumption | |
| *Desmodesmus* sp. MAS1 and *Heterochlorella* sp. MAS3 | Alginate beads | Fe(II) | Passive handling systems | 5% reduction in $CO_2$ emissions | 80% reduction in renewable energy reduction rate | [136] |
| | | | Active handling systems | 80% reduction in $CO_2$ emissions | Renewable energy reduction rate reduced by 9% | |
| *Chlorella* sp. (FACHB-31) | Water hyacinth leaf biochar pellets | Cd(II) | - | - | Removal efficiency: 91.1% (3 cycles) | [95] |
| *C. vulgaris* | Calcium alginate | Fe(II) | - | - | Removal efficiency: 3.56% reduction (5 cycles) | [114] |
| | | Mn(II) | - | | Removal efficiency: 4.32% reduction (5 cycles) | |
| | | Zn(II) | - | | Removal efficiency: 4.87% reduction (5 cycles) | |
| *Sargassum vulgare* | Calcium alginate beads | Fe(III) | - | - | Removal efficiency: 22% (5 cycles) | [104] |

## 5. Future Prospects

A life cycle assessment and cost analysis are used in Table 2 to explain how freshly immobilized algae remove heavy metals from the environment [79]. Combining hybrid technologies is beneficial in providing efficiency and performance of microalgae immobilization. Large-scale wastewater treatment methods may benefit from switching to the best microalgae–bacteria combinations for co-immobilization processes. The symbiosis between microalgae and fungus lowers the cost of culture by increasing biomass and streamlining the solid–liquid separation process [137]. Wang et al. [129] used a mycelium-particle flocculant to aid the co-immobilization of microalgae and fungi and found that a mutualistic symbiotic system of microalgae and fungi significantly increased the resistance to Cd(II) and had higher stability and higher adsorption efficiency (98.89%). It is an intriguing possibility to utilize genetically altered microalgae to cleanse wastewater and sediments from various locations, since it improves their capacity to bind to very low amounts of certain metals [138]. Therefore, improved biosorbents obtained by the genetic engineering of microalgae types can further enhance the selectivity of microalgae for metals, and the fixed-point design of the immobilization nodes of the carriers facilitates the optimization

of the efficiency of algal immobilization. Closed and open reactors for immobilized microalgae culture are the desirable choices for effective heavy metal removal, and the design and selection of the reactor operation mode are important for the optimization of efficient remediation systems in aqueous media [139].

The further development of biosorption technologies based on immobilized algae will require detailed life cycle analysis to assess environmental impacts, and the field scale analysis of algal immobilization may significantly advance the field and provide techno-economic insights [41]. Most of the current research on immobilized algae is restricted to the laboratory and has not been expanded to practical industrial applications on a large scale, so specific life cycle assessments are lacking. The desorption, regeneration, and reuse of biosorbents are additional processes that require pilot-scale testing.

## 6. Conclusions

The heavy metal contamination in polluted water poses a significant threat to living organisms. Due to the complexity, expense, and limitations of conventional wastewater treatment methods, bioalgal remediation is regarded as a cost-effective and environmentally friendly alternative. Biosorption, bioaccumulation, and biotransformation are utilized by microalgae to remove heavy metals from the environment. The mechanisms acting on the adsorption of heavy metals by algae include ion exchange, chelation/complexation, electrostatic interactions, and surface precipitation. The algae also present different treatment mechanisms in the face of different heavy metal ions. Commonly used for adsorption, encapsulation, entrapment, and self-immobilization, immobilized algal techniques can effectively improve the stability of heavy metals adsorption by algae. However, immobilization techniques are still susceptible to cell permeability, poor mechanical properties, easy deactivation, and instability. A thorough analysis of the algal immobilization procedure is provided in order to provide a cost-effective and applicable immobilization technique. This paper studies the effects of adsorbent content, initial heavy metal ion concentration and type, temperature, pH, contact time, heavy metals system, and algae and carrier type on heavy metals adsorption by immobilized algae in order to identify strategies for optimizing the performance of immobilized algae. Immobilized algae have environmental benefits, and their reusability reduces the cost of the adsorption process, according to a life cycle assessment and techno-economic analysis. In conclusion, future research on algal immobilization is recommended. Microbial co-immobilization strategies are effective solutions to improve the efficiency of algal immobilization; however, more in-depth life cycle assessments are needed.

**Author Contributions:** Conceptualization, Writing—original draft, Z.C.; Writing—review & editing, A.I.O., D.W.R., W.-D.O. and P.-S.Y. All authors have read and agreed to the published version of the manuscript.

**Funding:** This research received no external funding.

**Institutional Review Board Statement:** Not applicable.

**Informed Consent Statement:** Not applicable.

**Data Availability Statement:** Not applicable.

**Acknowledgments:** Ahmed I. Osman and David W. Rooney wish to acknowledge the support of The Bryden Centre project (Project ID VA5048), which was awarded by The European Union's INTERREG VA Programme, managed by the Special EU Programmes Body (SEUPB), with match funding provided by the Department for the Economy in Northern Ireland and the Department of Business, Enterprise and Innovation in the Republic of Ireland.

**Conflicts of Interest:** The authors declare no conflict of interest.

**Disclaimer:** The views and opinions expressed in this review do not necessarily reflect those of the European Commission or the Special EU Programmes Body (SEUPB).

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
