# Peer review of "Remediation of Heavy Metals in Polluted Water by Immobilized Algae: Current Applications and Future Perspectives"

_sustainability, doi:10.3390/su15065128_

Round 1

Reviewer 1 Report

The text that is the subject of this opinion is a review of the literature in the field of remediation of heavy metals by immobilised algae. There is a real problem of the high degree of water pollution in many regions, mainly industrial. Due to this fact, there is a need for such remediation. Therefore, the problem is significant and shows great practical potential, and the rapid development of science, especially biotechnology, allows us to hope for a solution.

The review of the current scientific literature in this field prepared by the authors is of great cognitive value, mainly for scientists. It has been prepared quite carefully. However, while reading the text, some imperfections that must be corrected and supplemented can find.

  1. The title suggests the usage of immobilised algae for the purification of wastewater. It seems to be a narrowing down because the same problem and its potential solution apply to leachate from landfill and dumps or potable water intakes. Maybe better instead wastewater write just polluted water?
  2. In L60-61, the authors write that "algal growth in aquatic systems is greener" for what is it greener?
  3. A sentence from L69-70 ("Currently, heavy metal removal technologies targeted at wastewater have yet to fully transition to algae remediation technologies") seems grammatically incomplete and misleading.
  4. On the list of heavy metals (L91-92), aluminium (Al) is listed twice (unnecessarily repeated). Aluminium does not meet the definition of heavy metal from L87 (aluminium density is 2.7 g/cm3) and is not classified as heavy metal in the literature.
  5. The chapter title in L206 should be on a new page.
  6. In L445, there is the phrase "sodium alginate" repeated twice.
  7. The sentence from L463-464 ("The magnetic field produced by the negative forces on the biochar surface may also increase this permeability") contains the incomprehensible suggestion that the magnetic field is produced by "negative forces". There are two questions: first: what are these "negative forces"? Second: is really the magnetic field produced by some force? (other than magnetic interactions). After carefully analysing the sources, it can be concluded that this sentence is a slightly modified sentence from [93]: "magnetic field created by the negative force on the biochar surface may increase the permeability of the cell wall". This sentence, in turn, comes from (Chen et al., 2017)), where to read: "the magnetic field produced by the negative charges on the surface of the biochar may protect the living cells". As it can see, it is not about "negative forces", But "negative charges". Nevertheless, the source of the magnetic field is not electric charges, and at most such moving charges can induce a magnetic field! Due to the controversial substantive content of the sentence from L463-464, the proposal is to delete the whole sentence.
  8. Table 2 is placed at the very beginning of chapter 4 before any text. It seems like it should be placed near the first reference to it until line L547 (and then in 552).
  9. Chapter 5 ("Conclusion") is too short, contains general information and should be extended. The statement from L588 is also unjustified: "this paper investigates (...)" - the text is a literature review and is not strictly an investigation nature (rather a research).

Regardless of the indicated comments, the text has big cognitive value. After the authors have applied appropriate additions and corrections, it will be a valuable compendium of the current state of the art in knowledge in using immobilised algae to remove heavy metals from water.

Author Response

Point 1: The title suggests the usage of immobilised algae for the purification of wastewater. It seems to be a narrowing down because the same problem and its potential solution apply to leachate from landfill and dumps or potable water intakes. Maybe better instead wastewater write just polluted water?

Response 1: Thank you for your suggestion. We have modified the title.

Point 2: In L60-61, the authors write that "algal growth in aquatic systems is greener" for what is it greener?

Response 2: Thank you for your comment. We added the relevant content in Line 62-64.

Point 3: A sentence from L69-70 ("Currently, heavy metal removal technologies targeted at wastewater have yet to fully transition to algae remediation technologies") seems grammatically incomplete and misleading.

Response 3: Thank you for your comment. We checked this sentence and made adjustments.

Point 4: On the list of heavy metals (L91-92), aluminium (Al) is listed twice (unnecessarily repeated). Aluminium does not meet the definition of heavy metal from L87 (aluminium density is 2.7 g/cm3) and is not classified as heavy metal in the literature.

Response 4: Thank you for your comment. We removed the duplicates and deleted 'Al'.

Point 5: The chapter title in L206 should be on a new page.

Response 5: Thank you for your suggestion. We have adjusted this title to the new page.

Point 6: In L445, there is the phrase "sodium alginate" repeated twice.

Response 6: Thank you for your comment. We deleted one of the sodium alginate.

Point 7: The sentence from L463-464 ("The magnetic field produced by the negative forces on the biochar surface may also increase this permeability") contains the incomprehensible suggestion that the magnetic field is produced by "negative forces". There are two questions: first: what are these "negative forces"? Second: is really the magnetic field produced by some force? (other than magnetic interactions). After carefully analysing the sources, it can be concluded that this sentence is a slightly modified sentence from [93]: "magnetic field created by the negative force on the biochar surface may increase the permeability of the cell wall". This sentence, in turn, comes from (Chen et al., 2017)), where to read: "the magnetic field produced by the negative charges on the surface of the biochar may protect the living cells". As it can see, it is not about "negative forces", But "negative charges". Nevertheless, the source of the magnetic field is not electric charges, and at most such moving charges can induce a magnetic field! Due to the controversial substantive content of the sentence from L463-464, the proposal is to delete the whole sentence.

Response 7: Thank you for your suggestion. We have removed this sentence.

Point 8: Table 2 is placed at the very beginning of chapter 4 before any text. It seems like it should be placed near the first reference to it until line L547 (and then in 552).

Response 8: Thank you for your suggestion. We have swapped the position of Table 2 with the content of the text.

Point 9: Chapter 5 ("Conclusion") is too short, contains general information and should be extended. The statement from L588 is also unjustified: "this paper investigates (...)" - the text is a literature review and is not strictly an investigation nature (rather a research).

Response 9: Thank you for your suggestion. We have expanded the conclusion.

Reviewer 2 Report

The present manuscript, sustainability-2264031, is a review report on the remediation of heavy metals in wastewater by immobilized algae. The topic is interested and well fits with the scope and aim of the journal. In the current time, the contamination of aquatic water bodies by heavy metals is one of the major concerns at the global level. In the last decades, a lot of scientific findings on the remediation of metal polluted environment by various physical-chemical and biological approaches have been developed. The present manuscript has scientific merit. I recommend that a minor revision is warranted. A more detailed review can be found in the specific comments below. I ask that the authors specifically address each of my comments in their responses.

Line 12: the line should be corrected as … “resulting in severe heavy metal contamination in the aquatic environment.

Line 91: p-group??? A 1-2 line description about the p-group must be mentioned.

Define acronyms/Abbreviations when they first appear; thereafter directly use them. Be consistent, please See; Line No. Line 89-92.

Line 128-129, Algae will produce  more azoic high molecular weight extracellular biopolymers (EPS) in response to heavy metal ions. Authors are advised to go through the literature and correct this statement with supporting references.

Line 139” the “but: should be changed to “however”

Few grammatical, punctuation, clarity, and typo errors should be corrected.

Line 173-175. Have you the supporting reference in your claimed.

Line: 583-585. It seems the sentence is incomplete, check for clarity.

Table legends, figure captions, and footnotes need improvement. All legends, captions, and footnotes should have enough description for a reader to understand the figure without having to refer back to the main text of the manuscript.

The references must be also in the format of the journal.
In addition, I marked my recommendation in the attached PDF as highlighted.

Author Response

Point 1: Line 12: the line should be corrected as … “resulting in severe heavy metal contamination in the aquatic environment.

Response 1: Thank you for your suggestion. We have modified it.

Point 2: Line 91: p-group??? A 1-2 line description about the p-group must be mentioned.

Response 2: Thank you for your comment. We added the relevant content in Line 118-119.

Point 3: Define acronyms/Abbreviations when they first appear; thereafter directly use them. Be consistent, please See; Line No. Line 89-92.

Response 3: Thank you for your comment. We have added detailed names of chemical elements.

Point 4: Line 128-129, Algae will produce  more azoic high molecular weight extracellular biopolymers (EPS) in response to heavy metal ions. Authors are advised to go through the literature and correct this statement with supporting references.

Response 4: Thank you for your suggestion. We have modified this sentence and added the reference.

Point 5: Line 139” the “but: should be changed to “however”

Response 5: Thank you for your suggestion. We have modified it.

Point 6: Few grammatical, punctuation, clarity, and typo errors should be corrected.

Response 6: Thank you for your suggestion. We have checked and improved the content and phrasing of the paper.

Point 7: Line 173-175. Have you the supporting reference in your claimed.

Response 7: Thank you for your suggestion. We have added the reference for it.

Point 8: Line: 583-585. It seems the sentence is incomplete, check for clarity.

Response 8: Thank you for your suggestion. We have modified it in Line 575-578.

Point 9: Table legends, figure captions, and footnotes need improvement. All legends, captions, and footnotes should have enough description for a reader to understand the figure without having to refer back to the main text of the manuscript.

Response 9: Thank you for your suggestion. We have expanded the caption of figures and tables.

Point 10: The references must be also in the format of the journal.

Response 10: Thank you for your suggestion. We have checked that the reference format is in line with the journal.

Reviewer 3 Report

The topic of the paper is interesting and relevant today. However, there are some comments to improve the manuscript.

Abstract should be revised and rewritten. Abstract should be included such information: Purpose. Methods. Findings. Practical value or Recommendation.

It would be appropriate to highlight the objectives of this review at the end of the Introduction in order to clearly structure the presentation of the review. For example, in this way:
“The review aims to …. To achieve this aim, the following tasks were set: ...”

It would also be appropriate to provide a separate sub-item of the methodological approach used to review and evaluate the mechanisms of bioremediation and immobilization of heavy metal ions in wastewater by algae.

Author Response

Point 1: Abstract should be revised and rewritten. Abstract should be included such information: Purpose. Methods. Findings. Practical value or Recommendation.

Response 1: Thank you for your suggestion. We have revised the abstract to add purpose, methods and findings, and value.

Point 2: It would be appropriate to highlight the objectives of this review at the end of the Introduction in order to clearly structure the presentation of the review. For example, in this way:

“The review aims to …. To achieve this aim, the following tasks were set: ...”

Response 2: Thank you for your comment. We added the relevant content in Line 82-90.

Point 3: It would also be appropriate to provide a separate sub-item of the methodological approach used to review and evaluate the mechanisms of bioremediation and immobilization of heavy metal ions in wastewater by algae.

Response 3: Thank you for your suggestion. We have provided a separate sub-item of the methodological approach in Line 92-106.